# Regulation of Gene Expression by m6Am RNA Modification

**DOI:** 10.3390/ijms24032277

**Published:** 2023-01-23

**Authors:** Bianca Cesaro, Marco Tarullo, Alessandro Fatica

**Affiliations:** 1Department of Biology and Biotechnology ‘Charles Darwin’, Sapienza University of Rome, 00165 Rome, Italy; 2Department of Anatomical, Histological, Forensic & Orthopedic Sciences, Section of Histology & Medical Embryology, Sapienza University of Rome, 00165 Rome, Italy

**Keywords:** m^6^Am, RNA modification, gene expression

## Abstract

The field of RNA modification, also referred to as “epitranscriptomics,” is gaining more and more interest from the scientific community. More than 160 chemical modifications have been identified in RNA molecules, but the functional significance of most of them still needs to be clarified. In this review, we discuss the role of N^6^,2′-O-dimethyladenosine (m^6^A_m_) in gene expression regulation. m^6^A_m_ is present in the first transcribed nucleotide close to the cap in many mRNAs and snRNAs in mammals and as internal modification in the snRNA U2. The writer and eraser proteins for these modifications have been recently identified and their deletions have been utilized to understand their contributions in gene expression regulation. While the role of U2 snRNA-m^6^A_m_ in splicing regulation has been reported by different independent studies, conflicting data were found for the role of cap-associated m^6^A_m_ in mRNA stability and translation. However, despite the open debate on the role of m^6^A_m_ in mRNA expression, the modulation of regulators produced promising results in cancer cells. We believe that the investigation on m^6^A_m_ will continue to yield relevant results in the future.

## 1. Introduction

The emerging “epitranscriptomics” field studies the impact of RNA chemical modifications on gene expression regulation. Among hundreds of RNA chemical modifications, the methylation in position N^6^ of adenine, which characterized both N^6^-methyladenosine (m^6^A_m_) and N^6^,2′-O-dimethyladenosine (m^6^A_m_), is better characterized in mRNA molecules (Figure 1) [1].

m^6^A is the most abundant internal chemical modification in mRNA. Several studies reported its relevant role in regulating different steps of the mRNA expression, including splicing, nuclear export, stability, and translation (reviewed in [1]). On the other hand, m^6^A_m_ is found in the first position adjacent to the 5′-end cap structure, which is constituted by a N^7^-methylguanosine (m^7^G) connected via a 5′ to 5′ triphosphate bond (m7GpppN; also referred to as cap 0), in many mammalian mRNAs [2]. m7GpppN is added cotranscriptionally by three sequentially enzymatic activities and is followed by one or two 2′-O-methylated nucleotides to produce cap1 (m7GpppN_m_) and cap2 structures (m7GpppN_m_N_m_), respectively [2]. When the first transcribed nucleotide is an adenine, this can be further modified cotranscriptionally at the N^6^ position to produce m^6^A_m_ [3]. Quantification studies revealed that the percentage of m^6^A_m_ in mRNA species varies according to different organisms and cell types, spanning from 10% to almost 50% [4]. Moreover, m^6^A_m_ profiling performed in human and mouse tissues showed that m^6^A_m_ levels vary greatly between different tissues [5]. In view of its vicinity with the mRNA cap-structure, it has been hypothesized that there is a role for m^6^A_m_ in regulating decapping, and eventually stability and translation. However, its role in gene expression is still controversial. In addition, m^6^A_m_ can be found as an internal modified nucleotide within the small nuclear RNAs (snRNAs) U2, where it has been shown to regulate pre-mRNA splicing.

In this review, we describe the methodologies to profile m^6^A_m_, the enzymes responsible for installing and removing m^6^A_m_ from RNA, and the impact of this RNA modification in gene expression regulation. Furthermore, we discuss the emerging roles of m^6^A_m_ regulators in tumorigenesis.

## 2. Methodologies for m^6^A_m_ Detection

As for other RNA modifications, global quantification of m^6^A_m_ levels can be obtained by liquid chromatography tandem mass spectrometry (LC–MS/MS) or thin-layer chromatography (TLC) on single-nucleotide digested RNAs [6,7]. Analysis can be restricted to mRNA species by affinity purification using poly-dT oligonucleotides followed by in vitro decapping. Nevertheless, one of the major challenges in RNA modification studies is the identification and mapping of the modified nucleotide within specific transcripts. High-throughput sequencing methods coupled with immunoprecipitation of fragmented RNAs with m^6^A-specific antibodies have been developed to identify m^6^A_m_ containing RNAs. Even if the anti-m^6^A antibodies cannot distinguish between m^6^A_m_ and m^6^A, m^6^A_m_ is limited to the first transcribed position in mRNAs, while m^6^A is enriched in internal regions of the RNA molecules within a specific consensus motif. Thus, sequencing methods utilized for m^6^A identification, such as MeRIP-seq (m6A-seq) and miCLIP, can be applied to m^6^A_m_ mapping with specific bioinformatics analysis [8,9] (Table 1). However, m^6^A sequencing methods do not efficiently capture m^6^A_m_-containing transcripts. Therein, specific protocols for the identification of m^6^A_m_ methylomes have been developed, each with its advantages and limitations (Table 1).

m6Am-Exo-Seq [10] utilizes a 5′ -> 3′ exonuclease to degrade uncapped RNAs after fragmentation, leaving only capped 5′-end fragments. These are decapped in vitro, to favor the m^6^A_m_ recognition by the m6A antibody, immunoprecipitated with an anti-m6A antibody and sequenced (Figure 2a). A similar method called m6A-crosslinking-exonuclease-sequencing (m6ACE-seq) allows the detection of both m^6^A_m_ and m^6^A modifications [11]. In the m6ACE-seq the m6A antibody, which recognizes both modifications, is crosslinked to RNA after fragmentation, thus protecting the fragments from digestion with 5′ -> 3′ exoribonuclease (Figure 2b). Sequencing of protected RNA, followed by a dedicated bioinformatics analysis, allows the mapping of 5′-end (m^6^A_m_) and internal (m^6^A) modifications. An additional methodology developed for specific m^6^A_m_ identification is the m6Am-seq [12] (Figure 2c), in which capped RNAs are immunoprecipitated with an m7G antibody after fragmentation and treated with recombinant FTO RNA demethylases, which removes the methyl group from the N^6^ position. Control and demethylated RNAs are then immunoprecipitated by an anti-m6A antibody. Comparison of the two samples, FTO-treated and FTO-untreated, allows the specific identification of m^6^A_m_ sites (Figure 2c). Recently, an antibody-free approach has been developed for the enrichment of m^6^A_m_ containing RNAs, CAPturAM [13]. However, it has not yet been applied to transcriptome-wide studies.

## 3. Regulators of m^6^A_m_ Levels

Levels of m^6^A_m_ are regulated by specific proteins that install and remove m^6^A_m_ modification from specific transcripts (Figure 3). Specific writer proteins modify mRNA and snRNA molecules, while the removal of m^6^A_m_ depends on a single eraser protein.

### 3.1. m^6^A_m_ Writers: PCIF1 and METTL4

PCIF1 (CAPAM)

The enzyme responsible for the methylation in the N^6^ position of the 2′-O-methyladenosine residue next to the m^7^G-cap of mRNAs and some small nuclear RNAs (snRNAs) was discovered from fractionated HeLa extract in the early 1970s [14]. By using in vitro modified mRNAs and purified germ or reticulocyte ribosomes, the authors also show that in their experimental conditions the presence of m7Gpppm^6^A_m_ had a small positive effect on ribosome binding [14]. Nevertheless, the gene coding for the modifying enzyme, known as “phosphorylated CTD-interacting factor 1” (PCIF1) or “cap-specific adenosine methyltransferase” (CAPAM), was only identified in 2019 by four independent groups [10,15,16,17]. PCIF1 contains an N-terminal WW domain that mediates interactions with the Ser5-phosphorylated carboxy-terminal domain (CTD) of the major RNA polymerase II (Pol II) subunit, which is present at transcription initiation, and a core region that contains the “helical” and “methyltransferase” (MTase) domains [14]. Like capping enzymes, which interact with Ser5-phosphorylated CTD, the WW domain provides coupling of the m^6^A_m_ modification with transcription initiation. The helical domain is specific for the PCIF1 protein and forms a positively charged groove that is thought to function as the RNA-binding surface. The MTase domain is highly homologous to that of class I MTases, which includes, among many, DNA and RNA SAM-dependent m^6^A methyltransferases. It is characterized by a Rossman fold catalytic domain containing a conserved sequence motif (NPPF). The specific recognition of the m7G cap occurs through a specific site (m7Gsite) located between the helical and MTase domains [15]. Recombinant PCIF1 protein alone is sufficient to methylate in vitro m7G capped mRNA [10,15,16,17]. It is the only methyltransferase responsible for N^6^-methylation of the 2′-O-methyladenosine nucleotide next to the m^7^G-cap of mRNA. Therein, its depletion produces the complete loss of m^6^A_m_ in mRNA. Surprisingly, PCIF1 deletion in mice has no effects on viability and fertility but produces reduced body weight [18]. Similarly, in the human HEK293T cell line, PCIF1 knockout did not produced any growth defect but altered cell proliferation under oxidative stress conditions [15]. PCIF1 methyltransferase is conserved only in vertebrates, and it is absent in yeasts, insects, and worms. However, in there is present in *Drosophila* a catalytically dead PCIF1 that is unable to methylate RNA, but it is still able to bind Ser5-phosporilated CTD [17]. Interestingly, in human cells it has been shown that PCIF1 expression inhibits transcriptional activation of reporter genes [19]. Thus, these results suggest a methyltransferase-independent role in the regulation of RNA Pol II activity by PCIF1 that has been conserved in fly.

METTL4

The spliceosomal small nuclear RNA (snRNA) U2 contains an internal 2′-O methylated adenine, at position 30 in human U2 snRNA, that is specifically N^6-^methylated by the METTL4 protein [20,21]. METTL4 belongs to MT-A70-like protein family containing a C-terminal MTase domain with the catalytic DPPW motif followed by a middle domain (MID) and a N-terminal domain (NTD) [22]. Interestingly, the catalytic site of METTL4 is very similar to the methyltransferase METTL3, which is responsible for the formation of m^6^A within mRNA molecules [1]. However, while METTL3 requires the METTL14 protein partner to form a positively charged groove for RNA binding, METTL4 works as a monomer and the function of METTL14 is played by its NTD domain [21]. Contrary to PCIF1, METTL4 is conserved during evolution and U2 snRNA appears to be its only RNA target in vivo [20,21,22]. In in vitro methylation assays, recombinant METTL4 has a preference for A_m_ over A for N^6^-methylation, within the CAAGUG sequence (A is the methylation site) found in U2 snRNA [19,20]. However, when overexpressed in cells, METTL4 can also modify A instead of A_m_ in mRNAs containing the consensus HMAGKD (H = A/C/U, M = A/C, K = G/U, D = A/G/U), which also includes the U2 snRNA target sequence [20]. Interestingly, in human cell lines METTL4 was also found to localize in mitochondria where it acts as an mtDNA m^6^A methyltransferase [23]. The depletion of METTL4 in the HEK293T cell line completely abolished m^6^A_m_ from U2 snRNA but did not alter cell viability [19,20]. However, METTL4 knockdown in mouse 3T3-L1 cells resulted in altered adipocyte differentiation [24], and the loss of METTL4 in *Drosophila* cells greatly impaired the proliferation rate [25]. These disparate results indicate cell-specific phenotypes.

### 3.2. m^6^A_m_ Eraser: FTO

FTO

FTO (fat mass and obesity-associated protein) belongs to the Fe(II)/α-ketoglutarate acid (α-KG)–dependent AlkB family of dioxygenases. Various in vivo studies reported that FTO can demethylate both cap m^6^A_m_ and internal m^6^A sites in mRNAs and snRNAs [26,27]. Notably, different cell types exhibit distinctive FTO cellular localization that greatly influence substrate demethylation by FTO [26]. Nuclear FTO preferentially demethylates cap-adjacent m^6^A_m_ in RNA Pol II-transcribed snRNAs, and internal m^6^A_m_ and m^6^A in the snRNAs U2 and U6, respectively. On the other hand, cytoplasmic FTO acts on cap-m^6^A_m_ and internal m^6^A acts on mRNAs [26]. The localization of FTO has been shown to be dependent on the cell cycle phase and to be regulated by casein kinase II-mediated phosphorylation [27,28]. Still, different cell types exhibit different FTO nuclear/cytoplasmic ratios [26]. FTO exhibits the same demethylation activity in vitro toward internal m^6^A and m^6^A_m_ positioned in the same RNA but is influenced by the sequence and tertiary structure of the RNA molecule [26,27]. Structural studies confirmed that the catalytic activity of FTO is mediated by the recognition of the methyl group in N^6^ position of adenine rather than the 2′-*O* methyl group of the ribose [27]. Therefore, the substrate specificity of FTO depends on its localization more than on the type of modification. Since cellular m^6^A levels are considerably higher than those of m^6^A_m_, many of the effects of FTO have been ascribed to the m^6^A demethylation activity [26].

## 4. Function of m^6^A_m_ in mRNA Expression

The development of transcriptome-wide methodologies for m^6^A_m_ mapping and the possibility to alter m^6^A_m_ levels by modulating m^6^A_m_ regulators expression allowed for the study of the specific contribution of m^6^A_m_ in gene expression regulation. However, the results have been controversial so far (Table 2).

### 4.1. The Role of m^6^A_m_ in mRNA Stability

The m^6^A_m_ modification was initially shown to promote the mRNA stability of highly expressed genes [29]. The knockdown of FTO in the HEK293T cell line and the mapping of m^6^A_m_ sites by miCLIP [9] indicated that m^6^A_m_ methylated mRNAs exhibited increased expression levels [29]. Direct transfection of synthetic mRNAs in HEK293T showed that m^6^A_m_ modified transcripts were more stable than their nonmodified counterparts, and the presence of m^6^A_m_ next to the cap decreased in vitro decapping by Dcp2 [29]. Furthermore, the enforced expression of a cytoplasmic FTO in HEK293T caused a significant decrease of m^6^A_m_ containing mRNA but did not affect m^6^A containing mRNA [29]. However, in a later publication the same group demonstrated that cytoplasmic FTO had a significant impact on m^6^A levels within mRNAs and their stability when localized in the cytoplasm [26]. Furthermore, a major problem in utilizing FTO modulation to assess m^6^A_m_ function is that the dual activity of FTO demethylase makes it difficult to discriminate between m^6^A- and m^6^A_m_-mediated effects. Later, the influence of cap-m^6^A_m_ on mRNA stability was analyzed in additional cell lines: 3T3-L1 (mouse embryonic cells), HeLa (cervical cancer) and JAWS II (mouse immortalized immature dendritic cells) by direct transfection of m^6^A_m_-modified mRNAs [30]. Interestingly, the initial observation in HEK293 of the positive contribution of m^6^A_m_ in mRNA stability was confirmed only in JAWS II. Thus, these results indicate that the effect of m^6^A_m_ on mRNA stability is strongly dependent on the cell type. Furthermore, it was also shown that mRNAs bearing a 2′-O methylated adenine (cap1), even if not methylated in N^6^, had increased stability. However, it was also shown that neither the 2′-O-methylation nor N^6^-methylation of adenosine influenced in vitro sensitivity to decapping by hDcp2, which, on the other hand, was strongly affected by the type of starting nucleotide [30]. It should be mentioned that Dcp2 is not the only decapping enzyme in human cells [33]. Thus, it is possible that the presence of m^6^A_m_ close to the cap might affect other decapping activities. Moreover, decapping activity in vivo is strongly influenced by RNA structure, RNA uridylation, and RNA binding proteins [34].

As PCIF1 is the only enzyme capable of modifying the adenine next to the cap, its deletion should have given clear results on the role of m^6^A_m_ in mRNA stability. However, even in this case, the results from various studies were not in agreement [10,15,16]. Analyses performed in mouse and human cell lines reported that PCIF1 knockout affected the stability of a subset of m^6^A_m_-modified mRNAs [16,18]. Particularly, PCIF1 knockout in three different mice tissues (brain, spleen, and testis), which resulted in complete loss of m^6^A_m_, produced deregulation of different transcripts, with a major impact on pseudogene expression in the testis [18]. However, while in testis transcripts beginning with an A were prevalently downregulated, in the other tissues the presence of an A close to cap did not discriminate between positive and negative variation of RNA levels [18]. An additional study performed in mice analyzed the dynamic regulation of m^6^A_m_ in the fat liver from animals on a high-fat diet [31]. The authors utilized m6A-seq methodology to map m^6^A_m_ sites in liver by subtracting m^6^A sites identified in mESC knockout for the mRNA m^6^A methyltransferase METTL3. Interestingly, they found that fat mice presented significant changes in m^6^A_m_ levels in mRNAs involved in metabolic and obesity-related processes [31]. Furthermore, by using RNA-seq to measure levels of modified mRNAs, they found a positive correlation between m^6^A_m_ levels and mRNA stability. These results suggest that the effect of m^6^A_m_ on mRNA stability might depend on tissue-specific factors. PCIF1 knockout in the HEK293T cell line confirmed that the deletion of m^6^A_m_ correlates with a decrease in mRNA expression [16]. However, by using SLAM-seq (Thiol SH-Linked Alkylation for the Metabolic sequencing of RNA) [35], a method that enables the detection of RNA synthesis and degradation kinetics, and miCLIP [9], which allows m^6^A mapping, they found an effect only on the stability of low-expressed m^6^A_m_-marked mRNAs [16]. To sum up, the deletion of PCIF1 in different model systems indicated that the m^6^A_m_ modification has a positive effect on mRNA stability. However, this did not always apply to all modified mRNAs. Moreover, the results depended on the experimental conditions and the cell type utilized for the analysis.

In contrast to the data showing a positive role for m^6^A_m_ in cap 1 on mRNA stability, other studies reported opposite results [10,15]. By combining m6Am-Exo-Seq, a method developed for m^6^A_m_ mapping, RNA-seq, to measure steady-state levels of mRNAs, and PRO-Seq (Precision nuclear run-on sequencing) [36], to assess nascent RNA levels, in human MEL624 melanoma and the HEK293T cell line deleted for PCIF1, they did not detect a direct effect of m^6^A_m_ on mRNA stability [10]. On the contrary, changes in RNA abundance upon PCIF1 deletion were due to variations in transcription [10]. An additional recent study that developed m6Am-seq, a specific sequencing method for m^6^A_m_ mapping (Figure 2), confirmed that PCIF1 was not required for the stability of m^6^A_m_-modified mRNAs [17]. These discrepancies might result from the different methods utilized in these studies for m^6^A_m_ mapping and measurement of transcription dynamics. In particular, the studies that found a positive role for PCIF1 in mRNA stability have utilized m^6^A mapping protocols instead of specific m^6^A_m_ mapping methodologies. Moreover, as mentioned before, the activity of decapping depends on several factors (e.g., 5′-end sequence, uridylation, RNA binding proteins), including the 2′-O-methylation status of the second transcribed nucleotide (cap2), which has been recently shown to inhibit mRNA decapping independently from hDCP2 [32].

Finally, m^6^A_m_ close to the cap in mRNAs was also shown to inhibit microRNA-mediated gene silencing, which also involved decapping [29]. However, the initial observation was not followed by mechanistic studies.

### 4.2. The Role of m^6^A_m_ in mRNA Translation

Considering that the cap structure in mRNAs plays a critical role in translation, various studies aimed at understanding the contribution of m^6^A_m_ in cap1 (m7Gpppm^6^A_m_) on protein production. However, as for the role of m^6^A_m_ in mRNA stability, these studies gave controversial results. The translation initiation factor eIF4E recognizes the mRNA cap structure to stimulate translation initiation. However, the eIF4E binding to cap is achieved by the specific contacts with the m^7^G and the three phosphate groups but not of the first transcribed nucleoside [37]. Analysis of eIF4E-cap binding affinity by EMSA (electrophoretic mobility shift assay) and competitive assays, which analyzed the displacement of the synthetic cap from eIF4E by capped transcripts, showed that the presence of m^6^A_m_ has a modest effect on eIF4E-cap interaction [15,32]. Different studies analyzed the effect of caps containing m^6^A_m_ modification by ribosome profiling and reporter assays. Ribosome profiling performed in PCIF1-deleted HEK293T cells did not produce significant changes in mRNA translational efficiency while, in the same study, it was reported to affect the stability of a subset of modified mRNAs [16]. Conversely, a different study, which utilized the transfection of reporter GFP-encoding mRNAs, found that m^7^G-m^6^A_m_-containing mRNAs were translated less efficiently [10]. This result was also confirmed in in vitro translation assay. Moreover, by combining quantitative proteomic with m6Am-Exo-Seq, it was reported that, in PCIF1-deleted cells, m^6^A_m_-modified mRNA that are not altered at the transcript level presented increased protein levels [10]. This was also confirmed across the human tissues where m^6^A_m_ modifications identified by m6A-seq were found to be negatively correlated with the protein level obtained from the Human Proteome Map database [5]. Thus, these results suggest a negative impact of m^6^A_m_ on protein synthesis. On the other hand, a different study, which combined ribosome profiling and RNA-seq in PCIF1 knockout HEK293T cells, reported a positive effect of m^6^A_m_ on mRNA translation but no effect on mRNA stability [15]. Finally, ribosome profiling analysis performed on PCIF1 knockout mice found no correlation between changes in translation rates and the presence of m^6^A_m_ in the cap1 of mRNAs [18].

These discrepancies might depend again on the different methods utilized for identifying m^6^A_m_-modified mRNAs or from additional mRNA features that might impact protein production, such as the 2′-O-methylation of the cap structure, which has been recently shown to influence translation in a cell-specific manner [32]. In a recent study, which utilized the transfection of in vitro transcribed luciferase reporter RNAs carrying different adenine modifications in cap 0, cap 1, and cap 2, it was observed that the effect of m^6^A modification in the cap structure was cell-specific [32]. In human lung carcinoma A549 cells, the presence of m^6^A in cap 0 (m7Gpppm^6^A) did not impact translation efficiency. However, the presence of conventional cap1 (m7Gpppm^6^A_m_) and cap 2 (m7Gpppm^6^A_m_N_m_) resulted in an increase in protein synthesis. Interestingly, protein production was also stimulated by single 2′-O-methylation in cap2 (m7Gpppm^6^AN_m_). In contrast, in mouse dendritic JAWS II cells the presence of m^6^A in cap 0 decreased translation efficiency compared to transcripts with cap 0 without m6A. Conversely, the presence of m^6^A_m_ resulted in an increase in protein production. However, the introduction of a single methyl group at the second transcribed nucleotide in transcripts starting with adenine (m7GppAN_m_ and m7Gpppm^6^AN_m_) produced a decrease of translation efficiency compared to canonical cap1 [32]. The contribution of m^6^A_m_ to the translation efficiency of transfected in vitro transcribed mRNAs was also analyzed in human THP1 leukemia cell lines and mouse 3T3-L1 embryonic fibroblasts [32]. In this case, the presence of m^6^A_m_, cap 1, decreased the translation in 3T3-L1 compared to the presence of only cap 0, while it had no effect on THP1 cells. Thus, the effect of m^6^A_m_ observed in A549 and JAWS II cells is lost in THP1 cells while it was the opposite in the 3T3-L1 cell line. The presence of cap2 (m7Gpppm^6^A_m_N_m_) gave, in both 3T3-L1 and 3T3-L1 cells, results consistent with JAWS II cells, with a negative effect on translation, while it was positive in A549.

These data indicate that the presence of m^6^A_m_ modification in the mRNA cap impacts translation in a cell-specific manner, and that its effects also depend on that of 2′-O-methylation modification in the second nucleotide of the cap-structure. However, there are no current methodologies that allow high-throughput mapping of 2′-O-methylation in the cap structure of mRNAs.

### 4.3. The Role of m^6^A_m_ in Splicing Regulation

Different RNA-pol II-transcribed snRNAs, such as U1 and U2, contain cap-adjacent m^6^A_m_ installed by PCIF1 [38], and can be removed by FTO in the nuclear compartment [38]. FTO knockout HEK293T cells exhibited increased exon inclusion but, in view of their different demethylating activity, it is still not clear if this can be ascribed to altered m^6^A_m_ levels in snRNAs. Moreover, splicing defects were never reported in PCIF1 knockout cells. The spliceosomal snRNA U2 also contains an internal m^6^A_m_ modification. The depletion of METTL4 did not alter U2 snRNA expression levels but rather, both in human and fly cells, led to altered splicing regulation [20,21,25]. Bioinformatics analysis performed in HEK293T cells showed that the lack of METTL4-mediated U2 snRNA modification resulted in an increase in splicing of retained introns and inclusion of exons [20]. Thus, the internal m^6^A_m_ modification appears to have a negative effect on the U2 snRNA function in splicing. Mechanistically, it was hypothesized that the lack of m^6^A_m_ in U2 snRNA might affect the recruitment of spliceosome components, such as U2AF, or splicing regulators. Indeed, it was also found that most of the affected introns are enriched for the GGGAGGG motif that is recognized by the splicing regulatory protein hnRNP H2 [20].

## 5. The Role of m^6^A_m_ in Cancer

The first indication of the role of m^6^A_m_ in cancer came from a functional RNAi screening performed in human bladder cancer cells and xenograft mice that identified PCIF1 as a novel tumor suppressor [39]. However, this was not followed by further functional studies. Instead, a later study performed in colorectal cancer (CRC) indicated an opposite role for this modification [40]. Knockdown of the RNA demethylases’ FTOs in CRC cell lines promoted staminality and resistance to chemotherapy drugs. This was related to a global increase of m^6^A_m_ levels but not of m^6^A in mRNAs, measured by LC-MS/MS [40]. Moreover, mapping of m^6^A peaks by m6A-seq upon FTO silencing did not reveal any changes in m^6^A sites within specific mRNAs. However, it should be considered that FTO downregulation could produce slight variation in m6A levels across genes, which cannot be precisely quantified by standard m6A-seq methodology. Nevertheless, the knockdown of PCIF1 partially rescued the phenotype of FTO downregulation, therein indicating that high m^6^A_m_ levels play an oncogenic role in CRC. Notably, in primary tumors, FTO protein was found predominantly in the nucleus in healthy adjacent tissue and in the initial precursor lesion of CRC while it translocated to the cytoplasm during infiltration in submucosa [39]. Thus, these data indicate that FTO might acquire specific cytoplasmic demethylation functions versus m^6^A_m_ in CRC. PCIF1 protein and m^6^A_m_ levels were also found to be upregulated in gastric cancer cases (GC) [41]. Furthermore, PCIF expression increased with increasing disease aggressiveness and correlated with a poor survival rate [41]. The knockdown of PCIF1 in GC cell lines produced a strong decrease in proliferation and invasion potential. More importantly, the oncogenic effect was also confirmed in GC patient-derived xenografts, where PCIF1 silencing decreased tumor volume and inhibited lung metastases [41]. Analysis of differential m^6^A_m_ methylated transcript upon PCIF1 silencing by m6A-seq identified TM9SF1 (Transmembrane Protein 9 Superfamily Member 1), a regulator of autophagy [42], as a relevant PCIF1 target mRNA. Interestingly, by using polysome profiling upon PCIF1 modulation, this study reported that PCIF1 specifically repressed TM9SF1 mRNA translation without affecting its stability and global mRNA translation. However, the same study, by using transfection of reporter GFP mRNAs containing m7G-A_m_ or m7G-m^6^A_m_ cap1, found that the presence of m^6^A_m_ modification had a negative effect on reporter mRNA translation [41]. Thus, in this latter case a general effect of m^6^A_m_ on mRNA translation was observed. The oncogenic role of PCIF1 was also confirmed by Pan-cancer analysis and PCIF1 RNA was found to be upregulated in most tumors compared to normal tissues [43]. However, recently, in gliomas a tumor suppressor role was again demonstrated for PCIF1 [44]. In this case, PCIF1 knockdown promoted the proliferation of primary glioma cells, glioma cell lines, and glioma xenograft mice. Furthermore, increased PCIF1 levels in glioma cell lines impaired proliferation and promoted apoptosis [44], while overexpression of PCIF1 in glioma cells injected into the brains of mice reduced the growth of the tumors and extended the survival rates of the animals [44]. However, the study did not analyze changes in m^6^A_m_ levels or identification of differentially m^6^A_m_-methylated mRNAs upon PCIF1 modulation.

## 6. Conclusions

Cap structure plays a crucial role in gene expression regulation by controlling mRNA stability and translation. The discovery of the methyltransferase responsible for the m6Am modification in the cap structure opened interesting perspectives on the possible role of this modification in regulating mRNA levels and protein production. However, a major problem in the epitranscriptomics field is the mapping and quantification of specific modification within the transcriptome. Most of the studies performed on m^6^A_m_ have utilized methodologies developed for internal m^6^A mapping followed by specific bioinformatics pipelines. Only a few studies have utilized specific protocols for m^6^A_m_ identification. These caused the lack of reproducibility and produced controversial results on the effect of m^6^A_m_ on gene expression. Hopefully, more sensitive methods for m^6^A_m_ detection will be developed to resolve the apparently contentious results. Furthermore, the modulation of m^6^A_m_ modification can result in cell-specific effects and, in transfection experiments using in vitro transcribed mRNAs, these effects are influenced by the 2′-O-methylation statuses of cap1 and cap2. In conclusion, the role of m^6^A_m_ in gene expression regulation needs to be clarified by further and more detailed investigation.

Interestingly, the deletion of m^6^A_m_ regulator proteins is well-tolerated in vivo but greatly affects the survival of different types of cancer. Thus, these data indicate that inhibitors against these proteins might have future applications in clinics.

## Figures and Tables

**Figure 1 ijms-24-02277-f001:**
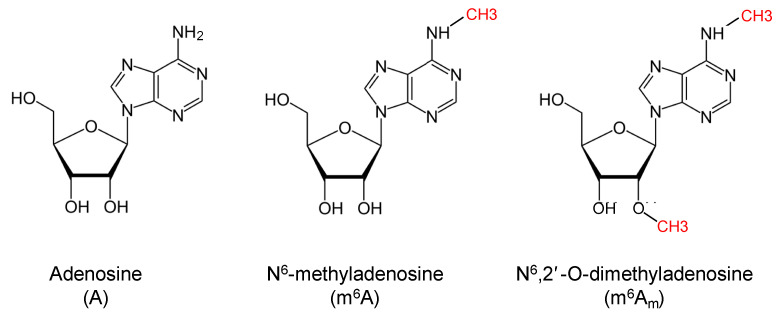
Chemical structures of adenosine (A), N^6^-methyladenosine (m^6^A), and N^6^,2′-O-dimethyladenosine (m^6^A_m_). Methyl groups are indicated in red.

**Figure 2 ijms-24-02277-f002:**
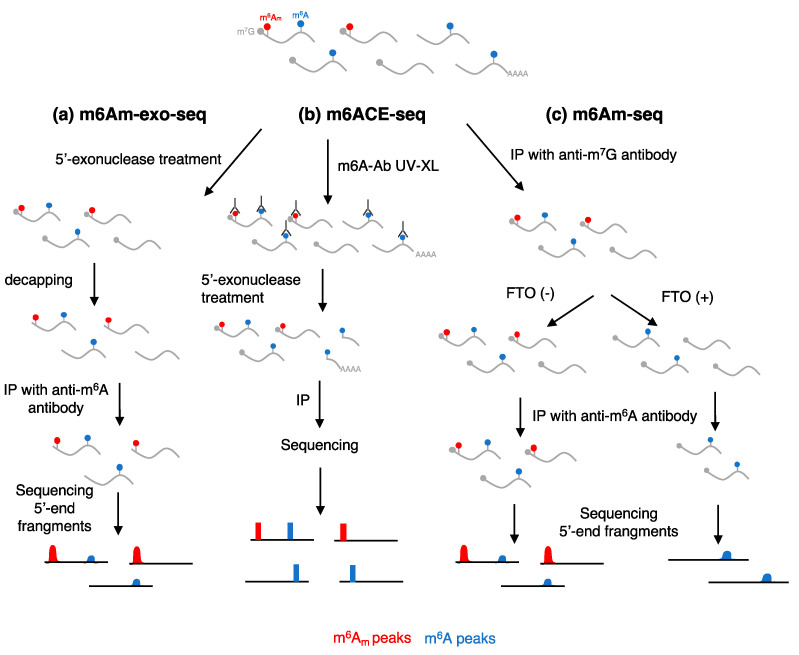
Sequencing methods for m^6^A_m_ mapping. All methods work on fragmented RNA. (**a**) In the m6a-exo-seq [10], RNAs are treated with a 5′-3′ exonuclease to enrich for m7G-capped RNAs. After decapping, fragments are immunoprecipitated with an anti-m6A antibody and sequenced. (**b**) In the m6ACE-seq [11], the anti-m6A antibody is covalently bound to m6A containing RNA fragments with UV crosslinking. Fragments are then treated with a 5′-3′ exonuclease, which is blocked by the bound antibody, to enrich modified RNAs before sequencing. (**c**) In the m6Am-seq method [12], m7G-capped fragments are immunoprecipitated with an anti m7G-antibody and then treated with FTO demethylases, which in vitro has better activity towards m6Am modification, before sequencing. FTO-untreated RNAs are utilized as control.

**Figure 3 ijms-24-02277-f003:**
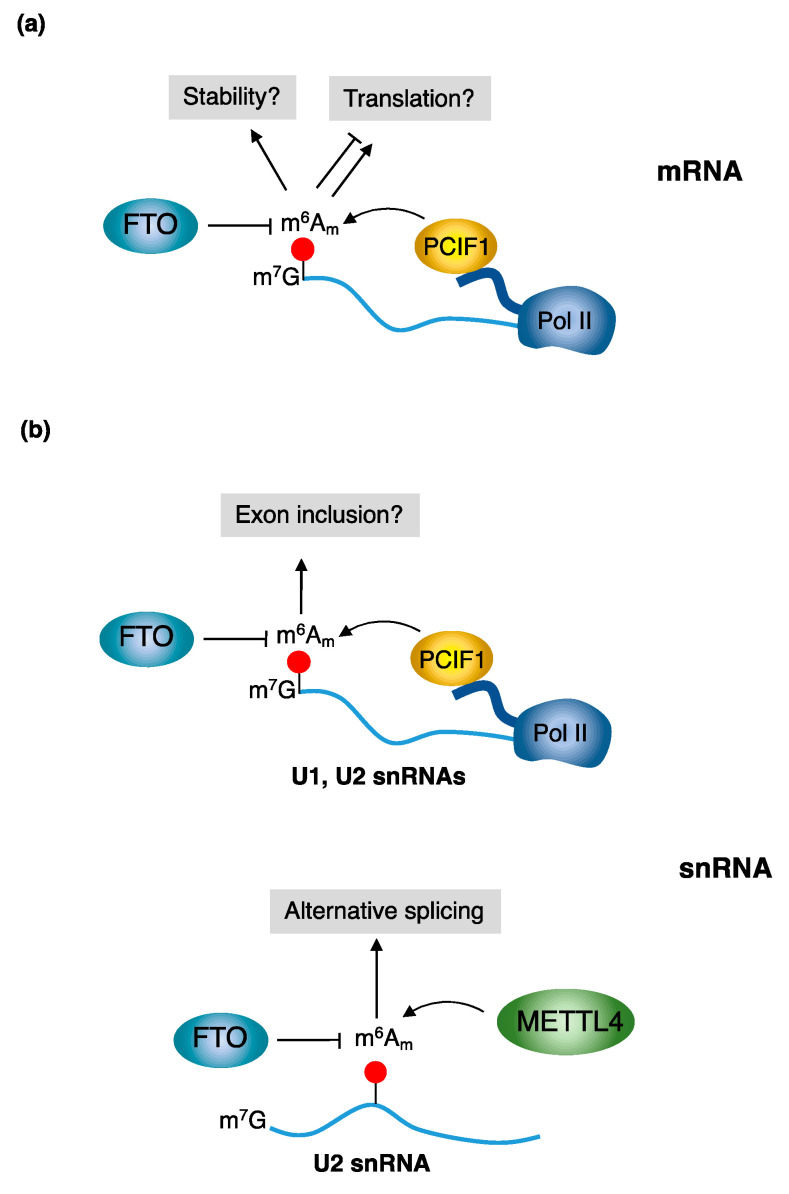
Regulators of m^6^A_m_ modification. (**a**) PCIF1 methylates the first transcribed 2′-O-methyladenosines, when present, producing m^6^A_m_ during primary mRNA transcription. This modification can be removed in the cytoplasm by FTO demethylases. The consequences of cap m^6^A_m_ deletion are still under debate. It was reported that m^6^A_m_ can increase mRNA translation, decrease mRNA translation, increase mRNA stability, and not affect mRNA stability (see main text for details). (**b**) Different snRNAs, including U1 and U2, contain m^6^A_m_ on their caps and this methylation is established by PCIF1 and removed by FTO. In addition, the U2 snRNA contains an internal m^6^A_m_ site that is installed by METTL4 and whose deletion causes altered alternative splicing.

**Table 1 ijms-24-02277-t001:** Methods utilized for m^6^A_m_ mapping.

Method	Principle	Advantages	Limitations
MeRIP-seq (m6A-seq) [8]	Utilizes anti-m6A antibody to enrich m^6^A- and m^6^A_m_-containing fragments	Easy to perform, kit available from different suppliers	Low resolution, required dedicated bioinformatic analysis for m^6^A_m_ identification, cannot distinguish between cap-m^6^A_m_ and m^6^A in 5′-RNA fragments, antibody cross-reactivity
miCLIP [9]	Utilizes CLIP with anti-m6A antibody to identify m^6^A- and m^6^A_m_ in RNA fragments	Single-nucleotide resolution	Required dedicated bioinformatic analysis for m^6^A_m_ identification, antibody cross-reactivity
m6am-exo-seq [10]	Utilizes 5′ -> 3′ digestion to degrade uncapped RNAs after fragmentation followed by m6A IP	Allows sequencing of cap-m6Am fragments	Cannot distinguish between cap-m^6^A_m_ and m^6^A in 5′-RNA fragments, antibody cross-reactivity
m6ACE-seq [11]	Utilizes crosslinking of anti-m6A antibody, followed by 5′-> 3′ digestion and m6A-IP to identify m^6^A- and m^6^A_m_ in RNA fragments	Single-nucleotide resolution, allows mapping of both m^6^A and cap-m^6^A_m_	Required dedicated bioinformatic analysis, antibody cross-reactivity
m6Am-seq [12]	Utilizes anti-m7G-antibody to purify 5′-RNA fragments, followed by digestion with recombinant FTO and m6A IP	Allows mapping of both m^6^A and cap-m^6^A_m_	Cannot distinguish between cap-m^6^A_m_ and m^6^A in 5′-RNA fragments, requires recombinant FTO protein, FTO activity is not specific for m^6^A_m_ and is influenced by sequence and structure, antibody cross-reactivity
CAPturAM [13]	Cap-m^6^A_m_ are enzymatically propargylated by PCIF1, using synthetic AdoMet analog, selectively biotinylated and enriched with magnetic streptavidin-beads	Antibody-independent method	Never applied to transcriptome studies

**Table 2 ijms-24-02277-t002:** Effects of m^6^A_m_ in mRNA stability and translation.

Enzyme/ Experimental Procedure	Molecular Effect of m^6^A_m_	Model System
FTO/KO [29]	Enhanced mRNA stability	HEK293T
FTO/OE [29]	Reduced mRNA stability	HEK293T
PCIF/KO [10,15,17]	No effect on mRNA stability [10,15,17]/increased translation [10]/reduced translation [15]	HEK293T, MEL624
PCIF1/KO [16]	Reduced stability of low expressed mRNAs/no effect on translation	HEK293T
PCIF1/KO [18]	Reduced stability of pseudogenes in testis/no effect on translation	Mouse
Transfection of in vitro transcribed RNAs [30]	Positive correlation with mRNA stability	JAWS II
Transfection of in vitro transcribed RNAs [30]	No effect on mRNA stability	HeLa, 3T3-L1
Sequencing [31]	Enhanced mRNA stability in fat liver	Mouse
Transfection of in vitro transcribed RNAs [32]	Positive effect of m^6^A_m_ on translation	A549, JAWS II
Transfection of in vitro transcribed RNAs [32]	Negative effect of m^6^A_m_ on translation	3T3-L1
Transfection of in vitro transcribed RNAs [32]	No effect on translation	THP1
Transfection of in vitro transcribed RNAs [10]	Negative effect of m^6^A_m_ on translation	HEK293T

KO: knockout, OE: overexpression.

## Data Availability

Not applicable.

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
