# Peer review of "Regulation of Gene Expression by m6Am RNA Modification"

_ijms, 2023, doi:10.3390/ijms24032277_

Round 1
Reviewer 1 Report
Review of IJMS 2156358
This manuscript by Cesaro et al provides a review of the current status of the m6Am RNA modification. Overall, this is a good, solid review that hits the important areas with respect to this interesting and functionally important modification, particularly the current state of its biological significance. The authors are particularly good at highlighting contradictory results, strengths and weaknesses of the papers that make these findings, and attempting to provide potential explanations.
There are some points that would improve the quality and readability of this review:
1. A bit more background to the review at the beginning would be helpful for the non-aficionados. Specifically, an introduction to both m6A and m6Am and the distinctions between them (locations and overview of roles) would be helpful, particularly since the sequencing techniques introduced in the next section rely heavily on distinguishing these modified nucleotides. Similarly, a schematic showing the chemical structures of unmodified A , m4A and the m6Am modification would be very helpful.
2. In the detection methodologies section, is there a potential role for Nanopore direct RNA sequencing, as there is for m6A?
3. Table 2- there are issues with spacing of the rows. This may have to do with how the table was set up in the original program, but needs to be addressed as it is confusion. For example, do text lines 4-6 in the molecular effect column refer to FTO/OE or to PCIF/KO in the Enzyme column and to which model systems in that column, etc. The spacing needs to be compensated (perhaps by empty lines or “cells” ) in the different sections to align properly.
4. The paper is in need of some grammatical/English usage corrections corrected by a native English speaker. For example, there are “sentences” without a subject/verb/object structure such as line 160 “Thus, indicating cell specific phenotypes” which is more appropriately the last clause of the preceding sentence and is not a sentence of its own. This can also perhaps be better stated as “These disparate results indicate cell-specific phenotypes”. There are a number of similar grammatical/usage levels issues.
a. The paragraph on PCIF1 (lines 220-242) is a bit confusing (probably since the results are confusing) and could use a summarizing sentence.
b. Other example, line 230, please clarify what is meant by “fat liver”- a liver from animal on high fat diet? A liver with NAFLD?
Author Response
- A bit more background to the review at the beginning would be helpful for the non-aficionados. Specifically, an introduction to both m6A and m6Am and the distinctions between them (locations and overview of roles) would be helpful, particularly since the sequencing techniques introduced in the next section rely heavily on distinguishing these modified nucleotides. Similarly, a schematic showing the chemical structures of unmodified A , m4A and the m6Am modification would be very helpful.
We thank the reviewer for the helpful comment. We added a background on m6A and m6Am in the introduction section. Moreover, we add a new figure 1 with the chemical structures of unmodified A, m6A and the m6Am.
- In the detection methodologies section, is there a potential role for Nanopore direct RNA sequencing, as there is for m6A?
Nanopore sequencing was never used for direct identification of cap-m6Am. The problem is that the quality of reads drops at both 5’- and 3’-ends, including adapters and part of the RNA sequence. Thus, the first 10 bases are clipped by the aligner during sequencing data analysis.
- Table 2- there are issues with spacing of the rows. This may have to do with how the table was set up in the original program, but needs to be addressed as it is confusion. For example, do text lines 4-6 in the molecular effect column refer to FTO/OE or to PCIF/KO in the Enzyme column and to which model systems in that column, etc. The spacing needs to be compensated (perhaps by empty lines or “cells” ) in the different sections to align properly.
We have modified the table according to reviewer comment.
- The paper is in need of some grammatical/English usage corrections corrected by a native English speaker. For example, there are “sentences” without a subject/verb/object structure such as line 160 “Thus, indicating cell specific phenotypes” which is more appropriately the last clause of the preceding sentence and is not a sentence of its own. This can also perhaps be better stated as “These disparate results indicate cell-specific phenotypes”. There are a number of similar grammatical/usage levels issues.
We apologize. We have corrected the text.
5. The paragraph on PCIF1 (lines 220-242) is a bit confusing (probably since the results are confusing) and could use a summarizing sentence.
We thank the reviewer for the comment. We added a summarizing sentence at the end of the paragraph.
6. Other example, line 230, please clarify what is meant by “fat liver”- a liver from animal on high fat diet? A liver with NAFLD?
We have specified in the text that it refers to animal on high fat diet
Reviewer 2 Report
In this review, Cesaro et al. systematacially described the methodologies to profile m6Am, the enzymes respon-37 sible for installing and removing m6Am from RNA, and the impact of this RNA modifica-38 tion in gene expression regulation. These contents were abundant, advanced, and they pointed out the future directions of relevant studies. I believe the manuscript was suitable for publishing in this journal as a Review Article.
Author Response
We thank the reviewer for the positive comments on our manuscript.
Reviewer 3 Report
This manuscript summarized the research progress on m6Am. The authors fully described the methodologies, writers/erasers, and functions of m6Am. This manuscript will be useful for readers in this field. Some suggestions are listed below, and hopefully, they can improve this manuscript.
1. The limitations/shortcomings of each method should be mentioned. A table that contains the advantages and limitations will be helpful.
2. More discussion or perspectives will be helpful to guild this field.
3. A "than" was missed after "stable" in line 198.
Author Response
1. The limitations/shortcomings of each method should be mentioned. A table that contains the advantages and limitations will be helpful.
We thank the reviewer for the suggestion. We added a new Table 1 containing advantages and limitations of m6Am sequencing methods.
2. More discussion or perspectives will be helpful to guild this field.
We thank the referee for the suggestion.
3. A "than" was missed after "stable" in line 198.
We thank the reviewer. We have modified the text.